



**The effect of tillage depth and traffic management on soil properties and root development during two growth stages of winter wheat (*Triticum aestivum L.*)**
**David Hobsonᵃ, Mary Hartyᵃ, Saoirse R. Tracyᵃ, Kevin McDonnellᵃ, ᵇ**
ᵃ *School of Agriculture and Food science, UCD, Belfield, Dublin 4, Ireland*
ᵇ *Biosystems Engineering Ltd, NovaUCD, Belfield, Dublin 4, Ireland*
*Correspondence to:* David Hobson (david.hobson@ucdconnect.ie).
**Abstract**
The management of agricultural soils during crop establishment can affect root development by changes to soil
structure. This paper assesses the influence of tillage depth (250 mm, 100 mm & zero) and traffic management
(conventional tyre pressure, low tyre pressure & no traffic) on wheat root system architecture during winter wheat
(*Triticum aestivum L.*) tillering and flowering growth stages (GS) on a long-term tillage trial site. The study
revealed that zero-tillage systems increased crop yield through significantly greater root biomass, root length
density and deeper seminal rooting analysed using X-ray Computed Tomography (CT). In general, conventional
pressure trafficking had a significant negative influence on crop yield, root development, bulk density and total
soil porosity of deep and shallow tillage conventional pressure systems compared no traffic zero and deep tillage
systems. Visual improvements in soil structure under zero tillage may have improved crop rooting in zero tillage
treatments through vertical pore fissures (biopores), enhancing water uptake during the crop flowering period.
This study highlights the implications of soil structural damage on root system architecture created by compaction
in crop production. The constricted root systems found in conventional pressure shallow tillage, zero and deep
tillage trafficked regimes emphasizes the importance of using technology to improve soil management and reduce
the trafficked areas of agricultural fields.
**1. Introduction**
Soil resources are under significant pressure from anthropogenic activities especially conventional tillage. The
resulting soil degradation has significant implications for food security globally (Lal, 2010). Changing weather
patterns from prolonged rain to drought periods are being experienced on a global scale, substantiating the
challenges faced by food producers. In 2018, worldwide wheat production fell by 34.5 million ton due to
prolonged droughts across Europe, Australia, and Canada. Soil compaction from field traffic is a well-recognized
problem in many parts of the world (Chan et al., 2006; Arvidsson and Keller, 2007; Naderi-Boldaji et al., 2018 )
affecting 33 million hectares in Europe alone (Akker and Canarache, 2001). Soil compaction is a form of physical
degradation caused by short crop rotations and heavy farm machinery working on low organic matter soils in wet
conditions resulting in the loss of pore space due to an externally applied load, forcing soil aggregates together





(Defossez and Richard, 2002). The resulting anaerobic high density soils have significantly reduced capacity to
store water and nutrients required by growing crops (Hamza and Anderson, 2005) and severely compacted soils
prevent soil exploration from root growth (Tracy et al., 2012).
Soil compaction is due in part to the pressure to complete field operations such as harvesting or drilling often in
short windows of good weather, which is exacerbated by the increasing use of larger machinery with increasing
axle loads designed to improve operational efficiencies. Common agricultural operations are conducted using
wheeled farm machinery which has tripled in weight and power since 1966 with wheel loads rising by a factor of
six (Chamen, 2006). When soils are cultivated in moist or wet conditions, soils can not withstand the compressive
forces applied post cultivation by heavy farm machinery traffic during operations such as seeding (Raper, 2005),
resulting in soil degradation (Batey, 2009). When soil is wet, tyre stress can propagate a greater distance down
through the soil profile. The depth and severity of soil stress is related to soil moisture, traction device applied
(track or tyre), track size, tyre inflation pressure and wheel load (Naderi-Boldaji et al., 2018).
Reforming the approach to soil management to mitigate challenges such as soil compaction and soil erosion offer
significant financial and environmental benefits compared to conventional agriculture. Cultivation practice using
minimal, or zero tillage techniques are widespread across many climatic conditions from semi-arid Canadian
plains to the temperate climates of Western Europe. In conventional tillage, the soil is either inverted >200 mm
using a mouldboard plough or deeply ripped using tines. The soil is then cultivated again to break down soil
aggregates to a crumb structure or fine tilth that is suitable to plant seeds (Morris et al., 2010). Conservation
tillage, also known as non-inversion tillage or reduced tillage, has been used for decades to improve soil structure
and health (Skaalsveen, Ingram and Clarke, 2019). Under conservation tillage, soil is disturbed to a lesser extent
(<100 mm using tines or discs) or not disturbed at all such as under zero tillage which involves the direct placement
of seed into undisturbed crop residues ( Soane et al., 2012).
The successful adaption of reduced tillage systems is not universally guaranteed with factors such as soil texture
and drainage, crop type and weather influencing successful implementation (Soane et al., 2012). In northern
Europe, crop yields under reduced cultivation systems rarely exceed those achieved by ploughing (Arvidsson,
2010). The exception under drier arid climates such as Spain, no tillage improved crop yields by moisture retention
in below average rainfall years ( Muñoz-Romero et al., 2010). Higher bulk density and penetration resistance are
typically found throughout the formerly tilled or "plough pan" layer in no tillage soils within the first two years
of adoption, resulting in root mechanical impedance (V. Boguzas et al., 2006). Yet, over time, long term zero
tillage has shown to attribute improvements in soil pore architecture and continuity throughout the soil profile by
bioturbation, suggesting roots could penetrate to lower soil horizons ( Cooper et al., 2021).
To date, studies have focused on how tillage influences physical soil properties (bulk density, cone penetrometer,
soil aeration) with root and crop yield responses (Whalley et al., 2008; Pires et al., 2017; Czyż, 2004). Soil types
and tillage systems have a considerable influence on the structural integrity of soil which controls rooting potential
(Morris et al., 2017). Studies have shown that low pressure tyres can reduce surface compaction compared to high
tyre pressure (Soane et al., 1980; Boguzas and Hakansson, 2001). As trafficking increases soil strength and
reduces a plant root's ability to penetrate soil layers, it is important to understand the relationship between tillage
depth and root system architecture during the growing season in response to trafficking. A dearth of information



exists on how tillage depth and tyre pressure affect rooting properties and crop yield on longer term field sites.
Yield reduction by soil surface compaction can increase abiotic stress in plants in three ways. It reduces soil
aeration, increases mechanical impedance of roots which in turn reduces root exploration of soil thus, mitigating
the extraction of water and nutrients from the soil resource (Chamen, 2011).
Quantitative measurement of root system architecture in three dimensions (3D) has become tractable using X-ray
CT in pot experiments (Mairhofer et al., 2017). Few examples of root studies using high resolution X-ray
computed tomography have been successfully conducted in field trials using undisturbed soil cores. Many studies
have focused on measuring soil structural properties such as porosity, soil pore size and distribution and the
influence of tillage method and trafficking (Millington et al., 2017; Rab et al., 2014). However, studying root
development and architecture in three-dimensional field structured soils remains challenging with X-ray CT due
to a bottleneck of rapid and standardized root extraction methods available, insufficient resolution and inability to
segment similarities in grey scale values between root and organic materials  (Zhou et al., 2021; Mooney et al.,
2012; Pfeifer et al., 2015).
The purpose of this paper was to identify the in-situ relationships between tillage depth and crop establishment
method on root architecture and crop yield under different traffic methods during two key growth stages of winter
wheat. X-ray CT was deployed to show if root architecture behaviors could be captured in-situ to the soil structural
environment created by the tillage method. Three cultivation practices and traffic management systems were
studied: Deep tillage (250 mm), shallow tillage (100 mm) and zero tillage, under no traffic, low tyre pressure and
conventional tyre pressure. The objectives of this study were to (i) assess the relationship between of traffic
management and three tillage depths and its effects on root system architecture and soil physical properties (ii)
Utilise 3D image analysis along with 2D destructive methods to verify rooting properties responsible for crop
yield.

## 2.  Materials and Methods

2.1 Site and soils
The study took place during the 2018/19 growing season. The experimental site is 3.12 ha, located at Harper
Adams University (HAU), Edgmond, Newport, England (52.779738 N, -2.426886 W). The HAU site is a loamy
sand soil consisting of the Olerton and Salwick series soils (Eutric Endogleyic Arenosol and Chromic Endostagnic
Luvisol respectively) (Millington et al., 2017). Further details of the soil properties are described in Table 1. To
highlight if any site variability existed across the site, soil properties were examined for fertility (pH and nutrient
levels), bulk density, soil strength and soil moisture. Particle size analysis (Gee and Or, 2002) was conducted to
determine soil texture classifications. The trial site was established in 2011 for previous studies with plots and
treatments carried out in the same location.
In the year prior to this study, it was necessary to plant a break crop (2017/18) as part of a standard crop rotation
to improve soil conditions and reduce diseases such as take all *(Gaeumannomyces graminis var.* tritici). A field
bean (*Vicia Fabia*) break crop was planted, and yields were assessed to ensure the trial site was uniform with no
underlying issues. Since the trial site began, the crop rotation has been first winter wheat (*Triticum aestivum L.*)





harvest in 2012 followed winter wheat in 2013, winter barley (*Hordeum vulgare L.*) 2014, winter barley 2015,
followed by a cover crop "TerraLife-N-Fixx" (DSV United Kingdom Ltd, 2015); Spring oats 2016, spring wheat
2017 and winter beans 2018. For this trial, winter wheat (*Triticum. aestivum L. cv.* Graham*)* was drilled early
October 2018 when the soil was dry, friable and soil temperatures >6 °C. The seeding rate was 250 seeds per m$^2$
and drilling took place on the 5$^{th}$ of October. This is in line with local normal farming practice.

**Table 1.** Description of the topsoil (0-300 mm) properties for Harper Adams University trial site, Shropshire, UK.

| Property | Units | |
|---|---|---|
| Location | Latitude | 52.779738 N |
| | Longitude | -2.426886 W |
| Soil type | Landis group* | Argillic brown earths, brown sands |
| | Landis series* | Salwick, Ollerton |
| | FAO | Luvisol & Arenosol |
| Sand (2000-65µm) | g g$^{-1}$ dry soil | 0.743 |
| Silt (63-2µm) | g g$^{-1}$ dry soil | 0.115 |
| Clay (<2µm) | g g$^{-1}$ dry soil | 0.143 |
| Texture | SSEW class | Loamy sand |
| Organic matter (LOI) | g g$^{-1}$ dry soil | 0.044 |

*Landis Soil guide (Cranfield University, 2021).
LOI, Loss of Ignition.


2.2 Experiment design

The experiment was a randomised 3 x 3 factorial arrangement of 9 treatments in four complete replicate blocks.
Each plot was 4 m wide x 84 m long with exception of block 4. Block 4 is 78.2 m long for operational reasons.
Tramlines were at a 90° angle to plots with 24 m spacing for fertilising and spraying operations throughout the
growing season. A split-plot design was used, half the plot (30 m) designated for sampling and the other half was
undisturbed for yield data collection. The half plot for sampling was sub-divided for the two sampling stages,
ensuring sampling did not occur near the same location as the previous sample. Cultivation for spring beans in
2017 was performed at three depths, 250 mm for deep tillage, 100 mm for shallow tillage and direct into stubble
for zero tillage. In the winter wheat trial, soil cores were collected at tillering (Growth stage (GS) 25) and the
flowering stage (GS 61-69) (Zadoks, Chang and Konzak, 1974) in July 2019.
Three commercial crop establishment systems were used consisting of three different tillage depths. The following
tillage treatments are denoted as: Treatment 1 =  Deep tine cultivator at 250 mm (DT) for deep tillage similar to
(Ren *et al.*, 2019), treatment 2 = shallow disc cultivation at 100 mm (ST) and treatment 3 = zero tillage using a
direct seed drill (ZT). In combination with the different tillage depths, three traffic regimes were used in this study



no traffic (NT), conventional tyre pressure (CP) and low tyre pressure (LP). Tillage depths were combined with
traffic management practices for the 9 treatments (DTNT, DTCP, DTLP, STNT, STCP, STLP, ZTNT, ZTCP &
ZTLP).

2.2.2 Tillage equipment and tyres
Primary cultivations in HAU involved a rigid tine and conical disc cultivator (Vaderstad Topdown) at 250 mm
depth to cut surface residues, loosen, mix, and consolidate the seedbed. The same implement was used for shallow
tillage treatments with tines adjusted upwards to reduce tillage depth (100 mm). A 290 hp Massey Fergusson 8480
with a track width of 2.1 m was used. Increased flexion AxioBib tyres were fitted IF 650/85 R38 179D TL on the
rear axle and (IF 600/70 R30 159D TL) at the front. A pneumatic disc seed drill (Vaderstad Spirit) was used to
sow the crop with 167 mm row spacing. The same drill was used to sow the zero tillage plots with the tines and
discs lifted to minimise disturbance (Kaczorowska-Dolowy et al., 2019).
For the tyre pressure treatment, the conventional tyre treatments were inflated to 1 bar for front and rear tyres
during cultivations. Low tyre pressure treatments and controlled traffic farming (CTF) plots operated on 0.7 bar
front and 0.8 bar on the rear axle. A front weight block of 540 kg was applied to the tractor for tillage primary
cultivation. All operations were performed under the same wheel-ways to keep traffic free zones for CTF plots.
During harvest, a Claas Dominator combine operated on a 4-m header, matching plot sizes (Smith, 2016). Crop
husbandry was carried out in accordance to the AHDB guidelines and soil fertility test analysis (AHDB, 2018).



2.3.1 Soil physical properties
Soil bulk density samples were also collected within the trafficked and non-trafficked area of the plot, to represent
the bulk density of the tillage treatments. Samples were replicated three times. Each core sample was 50 mm in
width and 300mm in length. An Eijkelkamp® soil corer was used to take bulk densities samples. Each bulk density
sample was taken within 0.5 m of the location of the soil cores taken for X-ray CT. The objective was to represent
the physical constraints (or lack of) for root growth in each plot examined. The method used in this study involved
splitting the bulk density sample into three 100 mm sections (0-100 mm, 100-200 mm and 200 – 300 mm) similar
to (Smith, 2016). The corer was opened in the field and split using a knife and ruler.
The core sections were stored in resealable bags and labelled before transporting to the laboratory for analysis.
Intact fresh soil cores were weighed prior to drying to record sample fresh weights. Samples were placed into an
oven at 105ºC for 24 h and reweighed to determine moisture % as per equation 1 and dry bulk density as per
equation 2 (Campbell and Henshall, 2000).

170           Moisture % = fresh weight(g) – dry weight (g) / dry weight(g) *100       Equation 1



Dry bulk density (Mg m$^{-3}$) = dry soil weight (Mg)/ soil volume (m$^{-3}$)

172                                                                    Equation 2.



2.3.2    Penetration resistance (PR)
Soil penetration resistance data were collected on each plot (in the wheel-ways and in the centre of the plot) down
to 450 mm with a depth increment of 25 mm between each recorded penetrometer reading. A cone penetrometer
(Data Field, Ukraine) was used, recording soil strength in kPa, the location and depth via built-in GPS device.
Only the PR samples were recorded at 450 mm to complete a reading on the data logger. It is also widely known
that roots penetrate past "tillage pans" ( Bengough et al., 2011) . Five penetrations were made both under and
between the wheel ways on each plot at GS 25 sampling to represent each treatment. PR was measured when soil
conditions were at field capacity to ensure accuracy of each reading.

2.3.3    Soil porosity analysis
Before soil porosity analysis on ImageJ software (version 1.52) (Schneider et al., 2012) could commence, an
image stack was created in VG Studio Max® for each scan. The contrast was adjusted to improve the uniformity
and visibility of the soil pores. The register object tool corrected scan discrepancies for soil core angle.
Straightening the scan allowed a cylindrical shape to be cropped and the tube edges and air space outside of the
soil core removed. This enabled soil data to be captured throughout the soil core. A new volume was selected and
extracted from the original. This created a separate cropped image volume to work from. The surface
determination tool in VG Studio Max® was used to threshold pore spaces within the solid matrix. The tool defines
the contour of objects, separating 3D data into regions, providing meaningful soil data (Borges de Oliveira et al.,
2016). The image was then inverted to remove the extracted variables from the image and highlighting the pore
spaces in the soil core. The processed image was exported as an *.TIFF image stack for further analysis using
ImageJ software.
Soil pore characteristics were measured using X-ray CT to establish information about the 3D soil environment
for root growth without disrupting the structural integrity of the soil core. The original grey-scale X-ray CT images
were analysed using ImageJ software. The scale was set for each dataset to define to spatial scale of the active
image. The unit of length was set in millimeters and the known distance was 0.045mm (45µm). Each scanned
core was cropped to remove the area outside of the soil column. The action of soil coring during sampling had the
effect of loosening the bottom 20 mm of the core, therefore 415 slices at the bottom of each scan were discarded
to remove the loosening effect due to the sampling process. The downward movement of the PVC pipe also caused
a smearing effect on the soil at the outside edge of the core and this area was also removed by cropping.
The processed image was 1220 x 1220 pixels in size. Applying the contrast enhancement filter helped normalize
all slices. The filter reduces the differences in pixel grey-level between slices known as beam hardening
(Wildenschild et al., 2002). The ImageJ Huang automatic threshold algorithms were used for each scan to create
binarized images and separate the air-filled pores from the background region. The binarized scans were de-
speckled twice to remove unwanted noise within each scanned image, improving analysis and accuracy of the




investigated pores. The Look Up Table (LUT) was inverted to change the white pores to black, ensuring analysis
calculated the air-filled pores and not the soil matrix. The resulting binary images were analysed using the Analyze
Particles tool which provided information for average pore size, total area and percentage porosity for each
individual image.

2.4.1   *Soil core sampling*
Field soil core size was chosen to capture as much root material growing in the field as possible while minimizing
the trade-off that exists with the X-ray CT technology between image resolution and core size (Mooney et al.,
2012; Zhou et al., 2021). The core dimensions were consistently 70 x 300 mm (diameter x depth) for each sample.
Soil cores were extracted from the field sites at GS 25 in February and again at GS 61 in June. Sampling was
carried out at GS 61 during wheat anthesis, when root growth is at its peak (Gregory et al., 1978). Due to high
moisture deficits in HAU (43 mm) during sampling at GS 61 in early July, the soil sample area was wetted with
2.5 L of water and allowed to infiltrate. This lubricated the soil, reduced soil fracturing, and allowed tube insertion
and soil core extraction to take place as smoothly as possible. Polyvinyl chloride (PVC) drainage pipes were cut
to size (70 x 300 mm) and these tubes were used to collect soil cores (as per Millington et al, 2017).
A single wheat plant sample was located at random in each plot. The selected plant was cut at the base of the
stem with a scissors and the above ground biomass discarded The PVC tube was placed (plant centred) directly
over the remaining plant stubble to maximise root system capture. Tubes were inserted into the soil using a mallet
in the crop rows in the centre of the plots between the wheel tracks (not trafficked by wheel) for untrafficked
samples for no traffic samples. A second core was taken in the wheel way for the tyre pressure treatments. A small
block of timber was used when hammering in the tube to protect tubes and soil cores from damage. A total of 72
samples were extracted on each sampling occasion and examined in this study. The PVC tubes were inserted into
the soil to a depth of 300 mm. The soil core was extracted carefully using a spade and the sample locations were
backfilled with soil. Following sampling, cores were sealed (top and bottom) using tape, labelled, and carefully
placed into boxes protected with bubble wrap. Cores were tightly packed and insulated to minimise movement
and drying of samples during transit to the laboratory for analysis. Samples were transferred to refrigerated storage
(<4°C) to prevent and reduce compositional changes to the soil through biological degradation.




*2.5.1 X-ray computed tomography (CT) – Root analysis*

Soil cores were transferred to the University College Dublin (UCD) X-ray CT facility at the Rosemount
Experimental Research Station at Belfield Campus, UCD, Ireland. The soil cores were scanned using a Phoenix®
v|tome|x M 240 kV scanner (GE Measurement and Control solution, Wunstorf, Germany). The v|tome|x M was



set at a voltage of 90 kV and current of 400 µA to optimize contrast between background soil and root material.
A voxel resolution of 45 µm was achieved by using the 'Multi Scan option' to scan in 4 segments. A total of 1800
projection images per section were taken at 200 m/s per image using the 'Fast Scan option', which has the default
values of an image averaging of 1 and 0 skip. No filters were used during scanning. The total scan time per core
was 24 minutes or 6 minutes per section. Once scanning was complete, the images were reconstructed using
Phoenix datos|x2 rec reconstruction software, the four scans were assembled into one 3D volume for the whole
core. Core samples were scanned within a week of the sampling date, the scanned core was 300 mm in length and
70 mm diameter. The software corrected movements during the scanning process and removed noise from scanned
images.
*2.5.2 X-ray CT root segmentation*

Image analysis for X-ray CT images was performed using the software VGStudioMax®, version 3.2 (Volume
Graphics GmbH, Heidelberg, Germany) to segment roots and soil porosity. Roots were segmented by setting seed
points and using selected threshold values in the "*Region grower"* that enabled fast and accurate selection of grey-
scale voxels (3D pixels) pertaining to root materials. The root system was extracted from the greyscale CT image
of soil using the VGStudioMax® semi-automated local adaptive thresholding "*Region Growing"* selection tool,
similar to (Tracy et al., 2013).  Root volumes were calculated by segmenting the root region of interest (ROI).
Once the roots were segmented from the image, erosion and dilation tool was selected at 1 pixel using the *Region*
*Growing* tool. Root system architecture parameters such as root vertical depth, root volume and root surface area
were measured from the segmented root systems. Root vertical depth was calculated on the Z axis in
VGStudioMax® from the length of a complete root from the base seed point.

*2.5.3 Destructive 2D root analysis*
After the soil cores were scanned, the soil and root material were separated by root washing gently with a water
jet hose. Two sets of sieves with a mesh size of 2 mm and 1 mm collected root material. Roots were washed and
soil material removed before the roots were placed into a sealed and labelled bag filled with water. The washed
root samples were placed into a freezer until scanning and analysis with WinRHIZO™ scanning and software
(version 2016a Regent Instruments, Canada) commenced. The root samples were thawed before scanning with
the WinRHIZO™ software. Large root stumps were removed from the sample prior to placing it inside the tray to
reduce root misrepresentation (Wang and Zhang, 2009). Roots were placed onto a clear transparent tray (30 cm x
20 cm) with water. A pair of plastic forceps were used to spread out root seminal and lateral roots. Images were
scanned at a resolution of 600 dpi (42 µm pixel size) with an Epson Perfection V800 scanning system. Root
images were measured for root length, root surface area, average root diameter and root volume for the total soil
core. This output was used to verify the 3D root outputs from VGStudioMax® (Flavel et al., 2017; Tracy et al.,
2012). The WinRHIZO™ software enabled rapid assessment of root parameters. It calculated the root volume by
determining the average root diameter and root length by pixel counting the 2D root image and then assuming the
root shape was cylindrical. The WinRHIZO™ used a skeletonization method for characterizing root systems
(Himmelbauer, Loiskandl and Kastanek, 2004). The software uses greyscale values in *.TIFF file format. The





output of the images was distinguished by global thresholding analyses for root diameter while root length was
validated by skeleton images. After WinRHIZO™ scanning, the roots were removed from the scanning tray using
forceps. The root samples were dried at 70°C for 24 hours and the root biomass samples were weighed.
*2.6 Soil Moisture Deficit Model*

Soil Moisture Deficit (SMD) was calculated based on the SMD hybrid model for Irish grassland (Schulte et al.,
2005). Rainfall, wind speed (m/s), sunshine hours, maximum and minimum temperature data were taken from the
nearest weather station located in Newport, Shropshire 6km from the site (Met office, 2019).

*2.7 Statistics*

Data from the scanned (destructive and non-destructive) images and root biomass were not normally distributed.
Non-normal data do not meet the assumptions underpinning ANOVA (Analysis of Variance); therefore, all data
underwent log transformation (in Microsoft Excel) before being exported to Minitab 18® where analysis of
variance (ANOVA) was performed to homogenize the variances of the compared means (Poorter and Garnier,
1996). For linear regression analysis, residuals of data were made to ensure that the assumptions of the analysis
were met (normal distribution, constant variance, etc). Normality was tested using the Anderson-Darling test in
Minitab 18®.

**3. Results**
3.1 *Growing conditions during crop season*
In 2018, crops were established at low soil moisture levels, which may have reduced soil compaction caused by
tillage operations across all site locations. From January to August (2019), 418.6 mm of rainfall was recorded at
HAU, 68 mm in total for January and February. Soil moisture deficits reached 66.2 mm in HAU (Supplementary
fig. S1) by early June 2019. High soil moisture deficits were recorded from early April to June, causing drought
stress during rapid growth periods (Met office, 2019).
3.2.1 *Soil properties – Bulk density & Penetrometer resistance*
The calculated probability (*P*-value) and standard error of the mean (SEM) from one-way ANOVA analysis is
given in Fig. 1 for bulk density presented for 0-100 mm, 100-200 mm, and 200-300 mm measurements. In the top
0-100 mm, bulk density was significantly higher in DTCP (1.66 Mg m⁻³) and STCP (1.44 Mg m⁻³) treatments
compared to ZTNT (0.994 Mg m⁻³) and DTNT (0.97 Mg m⁻³) (P<0.01). STNT (1.09 Mg m⁻³) was significantly
higher than ZTNT and DTNT and only significantly lower than DTCP. In the middle horizon (100-200 mm), a
significant interaction between trafficking treatment was found. Bulk density was significantly lower in DTNT




(1.07 Mg m$^{-3}$) compared to DTCP (1.63 Mg m$^{-3}$) and ZTCP (1.58 Mg m$^{-3}$) treatments (P<0.05). In the bottom
200-300 mm layer measured, no significant tillage x traffic interaction was found (P>0.05).

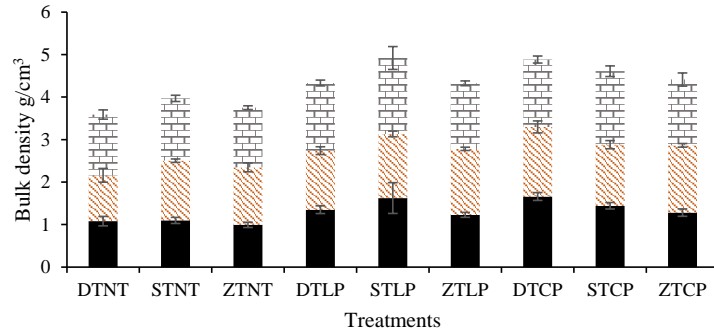

■ Bulk density 0-100 MM    ⧄ Bulk density 0-200 MM    ⊡ Bulk density 200-300 MM


**Figure 1.** Soil bulk density g/cm$^3$ for tillage x traffic treatments for three depth layers.
Penetration resistance (PR) was recorded in February 2019 when the soil was at field capacity. Measurements
were grouped into three groups, 0-150 mm, 150-300 mm, and 300-450 mm depth layers. Figure 2 depicts the
combined three layers grouped into one 0-450 mm graph. The ANOVA analysis revealed highly significant
differences for each layer. In the 0-150 mm layer, DTNT recorded the lowest kPa (kilopascals) readings and was
significantly lower than ZTCP, STCP, STLP, ZTLP and ZTNT (P< 0.000). DTCP and DTLP were significantly
lower kPa than ZTLP, STLP, STCP and ZTCP. ZTCP recorded the highest kPa reading and was significantly
higher than ZTLP, ZTNT, STNT, DTLP, DTCP and DTNT. In the second layer (150-300 mm), similar trends
were found and highly significant (P<0.000). STCP showed the highest kPa (3193.5 kPa) and was significantly
higher than STNT, ZTNT, DTNT, DTLP and DTCP. In contrast, DTNT recorded the lowest reading (1268.4 kPa)
and was significantly lower than ZTNT, STNT, ZTLP, ZTCP, STCP and STLP. STNT revealed significantly
lower kPa than STLP, ZTCP and STCP. ZTNT penetrometer readings were significantly lower than all trafficked
ZT and ST treatments. In the lower depth (300-450 mm), DTNT was significantly lower than STLP, STCP, ZTCP,
ZTLP and STNT (P<0.000).



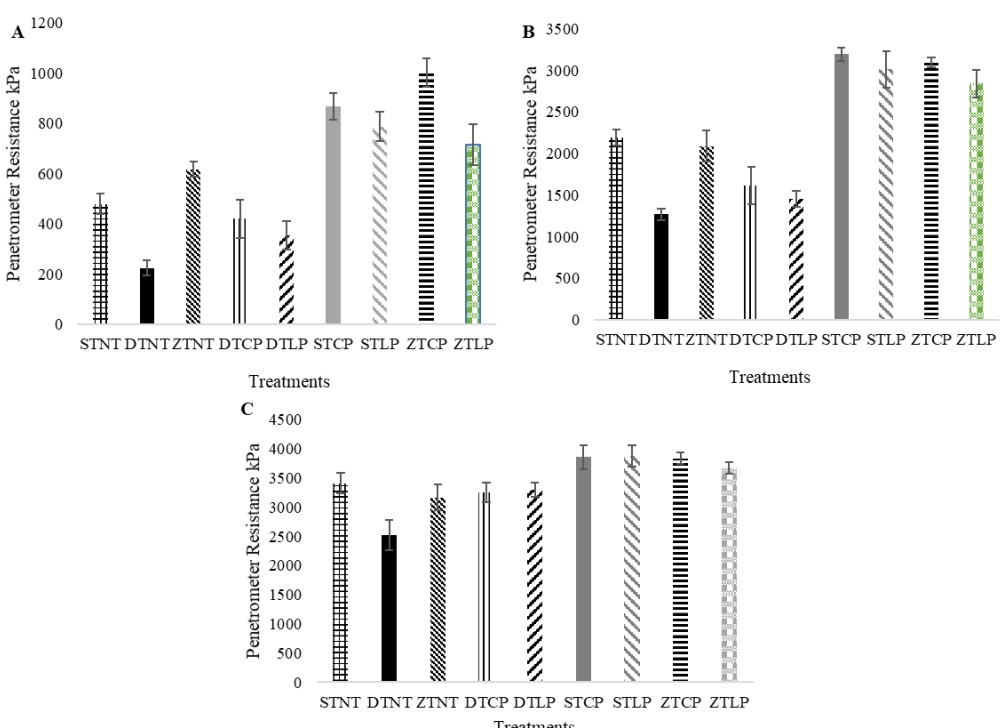


**Figure 2.** Penetration resistance for three layers (**a**) 0-150 mm (P<0.000), (**b**)150-300 mm (P<0.000) and (**c**) 300-
450 mm (P<0.000) during wheat tillering (GS25). Soil moisture conditions were at field capacity during sampling.

337

3.2.2 *Soil porosity*

The results of the ANOVA analysis of the CT-measured porosity (0-220 mm) are presented in Table 2. Soil
porosity results were split into two soil layers of 0-100 mm and 100-200 mm respectively. In the top 0-100 mm
layer, DTNT showed significantly higher total pore space (P<0.01) compared to all other treatments except ZTNT.
Tillage had a significant effect on soil porosity in the no traffic samples in the 0-100 mm layer (P< 0.05). Deep
tillage with no traffic had higher soil porosity (22.72%) than in shallow tillage (no traffic) (10.58%). There was
no significant difference between soil porosity under zero tillage and shallow tillage in the no traffic samples.
Trafficking had a significant effect on overall porosity. In deep tillage treatments, overall porosity 22.72% (no
traffic) was reduced to 8.08% (under low tyre pressure) and 6.50% under conventional tyre pressure. Traffic had
little effect on shallow and zero tillage porosity in the top 0-100 mm when compared to the no traffic samples
with small reductions in porosity. In the second examined layer, 100-200 mm zone, tillage and traffic were not
significantly different (P< 0.487). The percentage porosity shown in Table 2, indicate a sharp decline in the lower
depth with only 9.02% in DTNT. DTCP treatments recorded the lowest porosity (3.96%).






**Table 2.** Soil porosity for tillage x traffic for two soil layers.

| ImageJ soil porosity % 0-100mm | *n* | No traffic | low tyre pressure | Conventional tyre pressure |
|---|---|---|---|---|
| Deep | 4 | 22.72 **a** | 8.08 **b** | 6.50 **b** |
| Shallow | 4 | 10.58 **b** | 8.64 **b** | 7.23 **b** |
| Zero | 4 | 10.77 **ab** | 8.41 **b** | 8.49 **b** |
| P<0.01 | | | | |
| | | | | |
| ImageJ Soil porosity % 100-200mm | *n* | | | |
| Deep | 4 | 9.02 | 6.16 | 3.96 |
| Shallow | 4 | 4.06 | 6.44 | 5.32 |
| Zero | 4 | 2.895 | 6.44 | 5.32 |
| P<0.487 | | | | |

*Significant differences between means are represented by different letters.

3.3.1 *Destructive 2D root analysis*
The interaction between tillage system and trafficking protocols using destructive root measuring methods
(WinRHIZO™) are shown in fig 3 for GS 25 and fig 4 for GS 61. At GS25, no significant differences were found
between traffic and tillage treatments. However, the WinRHIZO™ analysis revealed a tendency towards increased
root growth in no traffic treatments. At the later growth stage (GS61), Figure 3 depicts the results showing highly
significant interactions between trafficking systems on root length density (RLD) (P<0.001) and root length (P<
0.001), root surface area (P< 0.002) and root volume (P< 0.05). DTNT showed significantly higher RLD, root
surface area and root length compared to ZTCP, STCP and STLP. Root volume was significantly higher in DTNT
over ZTCP and STCP. DTNT produced nearly double the root length compared to ZRCP. In contrast to DTCP,
root surface area reduced by 36% compared to untrafficked areas (no traffic samples). In shallow and zero tillage,
root surface area was reduced by 32% and 63.6% respectively in conventional pressure samples compared to
untrafficked samples. There was no significant difference for root diameter and between all tillage and trafficking
regimes. The results demonstrate that there was no significant difference in RLD at the tillering stage, nor could
trends be found as roots were undeveloped. However, at anthesis, the RLD was significantly higher under non-
trafficked tillage treatments when compared to DTCP, STCP and ZTCP (Fig 3b).




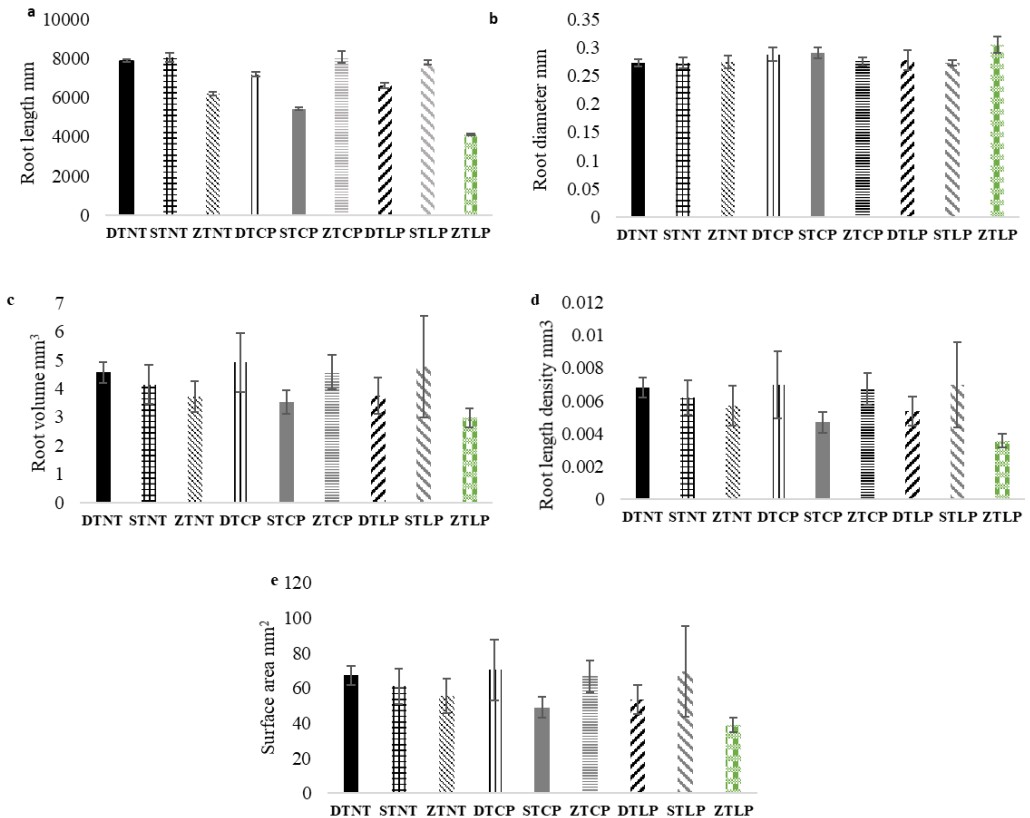


**Figure 3.** Tillering (GS25) root system architecture using destructive root method. (**a**) Root length (mm), (**b**) Root
diameter (mm) (**c**) Root volume (mm$^3$), (**d**) Root length density (mm$^3$), (**e**) Root surface area (mm$^2$).












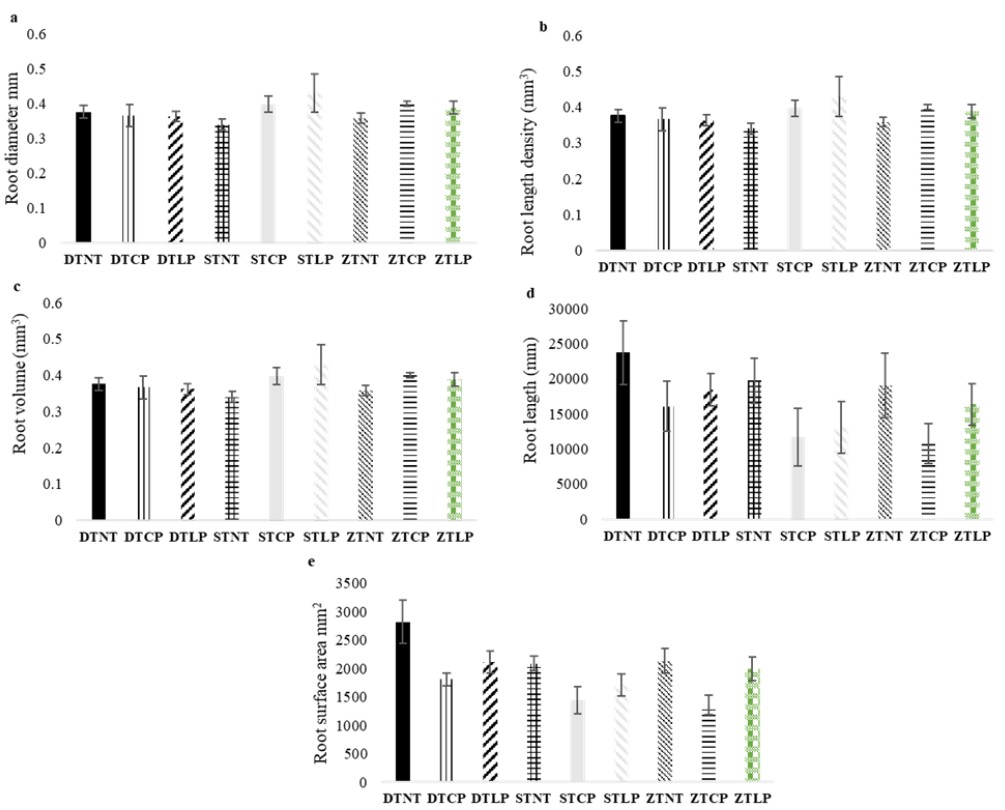

**Figure 4.** Flowering growth stage 61 root system architecture using destructive root method. (**a**) Root diameter, (**b**) Root length density (mm³), (**c**) Root volume (mm³), (**d**) root length (mm), (**e**) Root surface area (mm²)

3.3.2 *X-ray CT root analysis results*

Significant differences were found between trafficking treatments at GS61 for RLD and vertical root depth using non-destructive VGStudioMax 3.2 (Table 3). The X-ray CT scans revealed significantly longer vertical rooting (measured via the Z axis in VGStudioMax®) in ZTNT (112.7 mm) compared to DTCP (60.44 mm), DTLP (66.96 mm), STLP (65.39 mm) treatments (P<0.001). ZTNT showed significantly greater RLD (0.000098 mm/m³) over DTCP (0.000052 mm/m³), DTLP (0.000058 mm/m³), STLP (0.000058 mm/m³) and ZTCP (0.000060 mm/m³) treatments (P<0.001). Root volume and surface area showed no significant difference using X-ray CT. However, similar trends were found to the conventional WinRHIZO™ method. Trafficking had more of an influence on rooting than tillage method which did not have any significant effect on root parameters. As RLD is an important root trait commonly measured to estimate water uptake (White, Sylvester-Bradley and Berry, 2015), linear regression was used to verify the relationship between root depth and RLD. A significant relationship (P < 0.001) was found with a coefficient of determination $R^2 = 0.54$ (Supplementary Fig. S2).








**Table 3.** Root system architecture using non-destructive method.

| | Root system Architecture | | | |
| | flowering growth stage | | | |
| | Root surface area | | | |
| Tillage x traffic | Root volume mm3 | mm2 | Length (Z) axis (mm3) | Root length density (mm/m3) |
|---|---|---|---|---|
| DTNT | 3900.00 | 23448 | 96.1 **ab** | 0.000083 **ab** |
| STNT | 2648.00 | 17350 | 88.4 **abc** | 0.000077 **ab** |
| ZTNT | 3048.00 | 17907 | 112.7 **a** | 0.000098 **a** |
| DTCP | 2276.00 | 12114 | 60.44 **c** | 0.000052 **b** |
| DTLP | 3525.00 | 20269 | 66.96 **bc** | 0.000058 **b** |
| STCP | 2900.00 | 18052 | 67 **abc** | 0.000058 **ab** |
| STLP | 2358.00 | 14211 | 65.39 **bc** | 0.000057 **b** |
| ZTCP | 2533.00 | 15040 | 69.43 **abc** | 0.000060 **b** |
| ZTLP | 4480.00 | 25104 | 97.89 **ab** | 0.000085 **ab** |
| **P value** | NS | NS | 0.001 | 0.001 |

*Significant differences between means are represented by different letters.
Figure 5 shows root biomass results for GS25 and GS61. No significant differences between treatments at GS25
(P<0.848) were found. However, root biomass was significantly different for tillage x traffic with high confidence
level (P<0.001) at GS61. DTNT (0.829 g) showed significantly (P<0.001) greater root biomass, than STCP (0.437
g) and ZTCP (0.4530 g) treatments. DTNT did not significantly differ from ZTLP (0.7992 g), ZTNT (0.7939 g),
DTLP (0.6837 g), STNT (0.4991 g) and STLP (0.4923 g). The results show that, DTNT, ZTLP and ZTNT resulted
in nearly 50% greater root biomass over STCP and ZTCP treatments. Tillage treatments (center line where there
was no traffic effect) did not differ significantly with respect to root biomass.



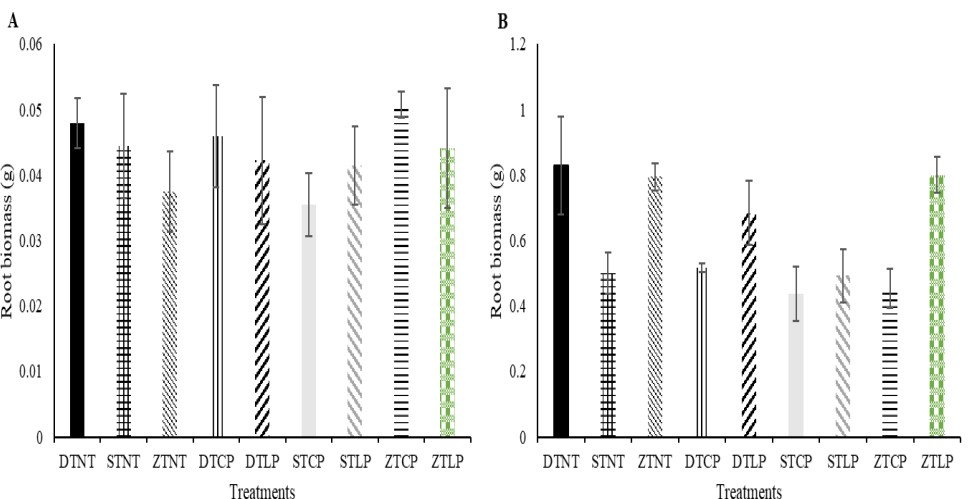


**Figure 5.** Root biomass at tillering (GS25) and flowering (GS61) for traffic and tillage treatments. Treatments
represented by initials (Tillage: D = Deep, S = Shallow, Z = Zero), (Traffic: NT = No traffic, LP = Low pressure
tyre, CP = Conventional pressure tyre).


3.4 *Crop yield*
Crop yield was highly significant between trafficking treatments and tillage (P<0.01) shown in Fig. 6. ZTLP had
the highest yield (11,385 kg ha$^{-1}$) and was significantly greater than DTLP (10,757 kg ha$^{-1}$), STCP (10,700 kg ha$^{-1}$
$^{1}$), STNT (10,678 kg ha$^{-1}$), STLP (10,638 kg ha$^{-1}$) and DTCP (10,613 kg ha$^{-1}$). All three zero tillage treatments
trended higher than deep tillage and shallow tillage treatments. ZTLP showed a 500 kg ha$^{-1}$ yield advantage over
DTNT (NS) and between 628 - 772 kg ha$^{-1}$ over trafficked treatments and STNT with high significance. In general,
this study did not show a trend in yield between conventional and low tyre pressure treatments. For deep tillage,
conventional tyre pressure reduced crop yield compared to low tyre pressure by 144 kg ha$^{-1}$ (1.34%). When
compared to the no traffic sample, conventional tyre pressure consistently reduced yield by 272 kg ha$^{-1}$ (2.5%)
in deep tillage. Although not significant, trafficking trended towards improving yield by 30 kg ha$^{-1}$ (0.03%) using
conventional tyre pressure and 340 kg ha$^{-1}$ (3.07%) using low tyre pressure. No trends were found in shallow
tillage treatments. Linear regression of root depth using X-ray CT showed a significant relationship to crop yield
(P < 0.001) and positive correlation (r = 0.54). However, the coefficient of determination was low $R^2$ = 0.3094
(Fig. S3). Moreover, regression analysis also showed a significant relationship between root biomass and crop
yield (P < 0.01). However, the correlation between the two variables was weaker (r = 0.43) (coefficient of variance
$R^2$ = 0.1859. This indicates that root depth is a stronger predictor of crop yield.






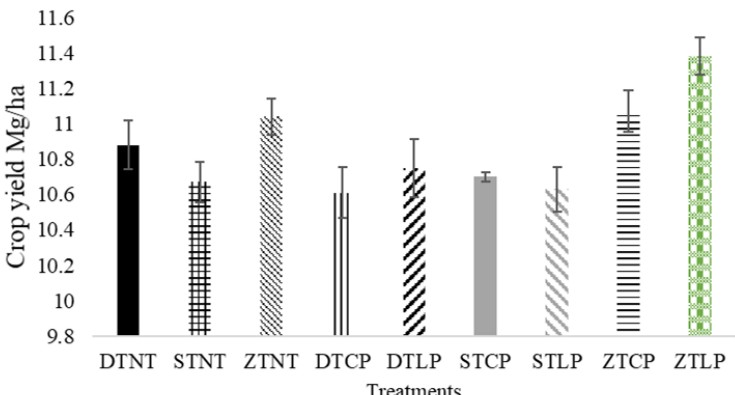


**Figure 6.** Crop yield in Mg/ha for traffic x tillage treatments.

**4. Discussion**
4.1.1 *Soil physical responses to tillage & trafficking*
In line with this papers hypothesis, trafficking effects were more influential on crop and root performance than
tillage system. The presence of wheeled areas in both zero and deep cultivation treatments increased soil bulk
density significantly in deep tillage treatments (Fig. 1). Previous studies have shown that zero tillage systems
increase in bulk density, penetration resistance and reduce in porosity in the early years of adoption from
conventional tillage systems (Christian and Ball, 1994; Six et al., 2004; Mangalassery et al.,2014a; Smith, 2016).
Vogeler et al., (2009) showed that bulk density is higher under conservation tillage methods in the top 100 mm
layer during the first five years of adoption from conventional systems. Indeed, Soane et al., (2012) reported that
significant regeneration of soil structure requires a three-year period from tillage depending on previous historic
land management practice. Moreover, values decrease in the long term with multiple benefits including improved
saturated conductivity, soil organic matter and air permeability in lower soil horizons. Arvidsson, 1998 showed
that soils with <30 g kg$^{-1}$ of organic matter were likely to suffer 11% higher crop yield loss due to compaction
using uniaxial compression tests. It is plausible that the actions of soil fauna such as earthworms and old root
channels could have reduced bulk density over time (Fig. 7) as identified by (Angers and Caron, 1998). Roots
promote soil structural formation through increasing soil aggregation. Root mucilage production, root hair
formation, and localised wetting and drying cycles encourage a reduction in soil bulk density (Bengough, 2012).
Our data shows similar findings with zero and deep tillage significantly reduced bulk density values in
untrafficked zones. However, in trafficked treatments, high tyre pressure combined with deep tillage treatments
resulted in higher bulk density values due to the loss of inherent strength by tilled soil, resulting in compression
of soil particles (Raper, 2005; Soane, Godwin and Spoor, 1986). Chan et al., (2006) observed that trafficking after
deep tillage increased bulk density values from 1.27 Mg m$^{-3}$ to 1.54 Mg m$^{-3}$, emphasizing the effect of trafficking
on the reduced bearing capacity of the deep tilled soil. The optimum soil density has been reported to differ



between soil types in previous studies. Indeed, Czyż, (2004) established a soil type interaction between crop yield,
bulk density and root mass concluding with sandy loam soils (similar to this study) having an optimum bulk
density value of 1.54-1.66 Mg m$^{-3}$. Yet, in this study, root biomass was significantly reduced with treatments
displaying similar soil density values to that reported optimum. Although conventional pressure tyres significantly
affected zero tillage in the 100 – 200 mm layer, trafficking affected the 0 – 200 mm later under deep tillage. In
shallow tillage treatments, the top 0- 100 mm layer was considerably impacted by high tyre pressure.

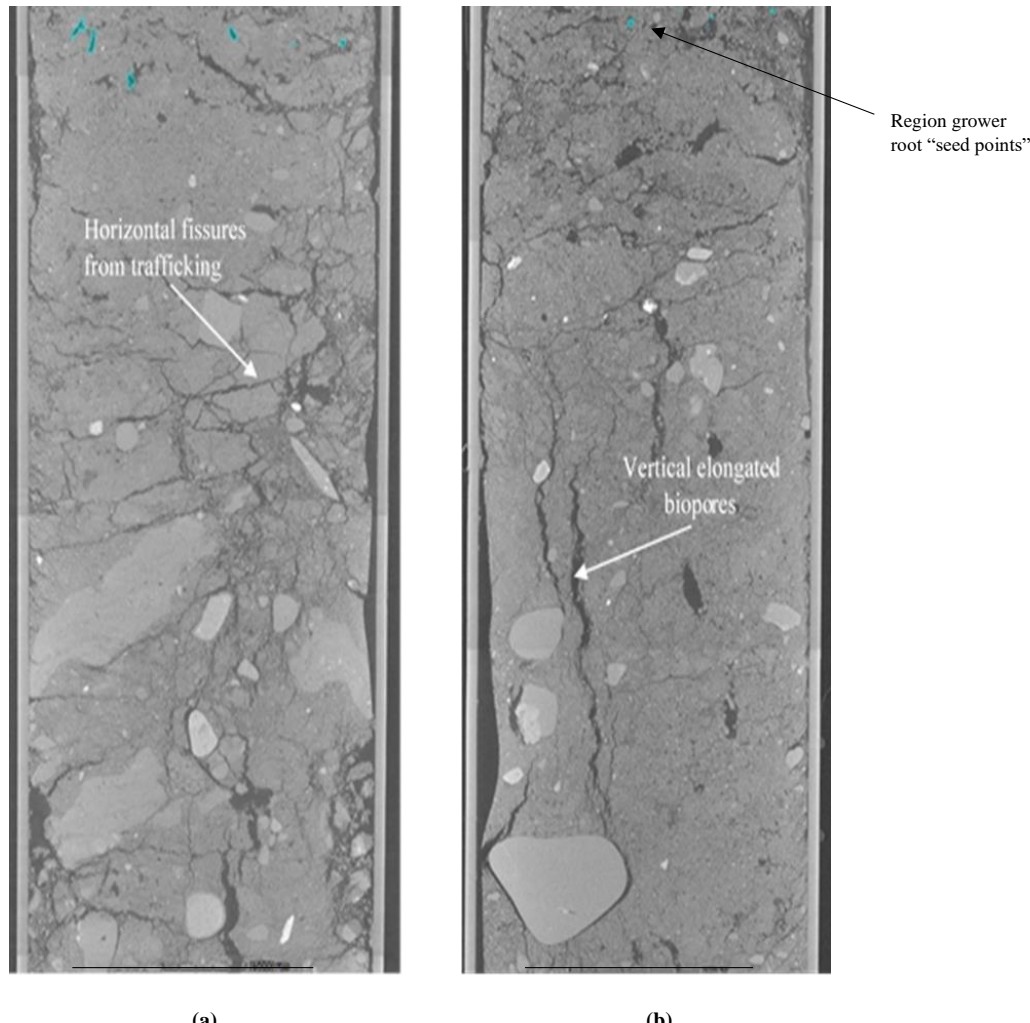


**(a)** **(b)**
**Figure 7.** Vertical view of X-ray CT images through centre of soil core using VGStudioMax® software for (a)
Shallow tillage conventional pressure (b) Zero tillage low tyre pressure. Scale bar = 50 mm.




4.1.2 *Soil porosity in response to trafficking & tillage*

Sandy soils due to their adhesive and coarse grain nature, have reduced porosity, including lower levels of
micropores compared to loamy soils (Arvidsson, 1998). The aggregation potential in this sandy loam soil is low.
In the presence of plants, porosity and pore connectivity as shown to reduce further compared to clay cohesive
soils which tend to increase in porosity through flocculation and aggregation (Bacq-Labreuil et al., 2018). Here,
we found soil porosity to be low in general across all treatments. When comparing cultivation systems, we found
that shallow tillage in the 0-100 mm layer had significantly lower porosity (10.58%) compared to deep tillage
(22.72%). Although zero tillage recorded low porosity values also (10.72%), it was not significantly different to
the other two systems. Compared to non-trafficked treatments, trafficked soil in general caused a sharp decline in
soil porosity in the top 0-100 mm layer. Tyre inflation pressure is one of the key contributors to soil stress in the
100 to 1000 mm layer (Botta et al., 2008). The effect of re-compaction from trafficking after cultivation was often
worse in deep tillage treatments, with a lower percentage porosity than in zero and shallow tillage (Table 2 for
DTLP and DTCP treatments). In deeply cultivated soils, water infiltration rates can be reduced by up to 82% after
a single wheelings (Chyba, 2012), which has agronomic implications such as reduced water and nutrient use
efficiency by up to 22% thus, potentially resulting in crop yield penalties of up to 38% ( Ishaq et al., 2001) . Yield
effects by trafficking were modest in our study due to low soil moisture conditions during sowing in autumn 2018
(Met office, 2019). Dry soil has increased soil strength, reducing the effects of soil compaction as the soil load
support capacity would have increased thus, increasing permissible ground pressure (Hamza and Anderson, 2005).
A key characteristic of zero tilled soils is a change in soil pore architecture with vertically orientated fissures
connected down through the soil profile created by biopores (Fig. 7). Similar findings have resulted in reduced
$CO_2$ fluxes and increased saturated hydraulic conductivity by surface-connected porosity (Cooper et al., 2021).
The same study found similar soil porosity levels between conventional and zero tillage with zero tillage total
porosity ranging from <5%, 10% and 12% on average over 1-5, 6-10 and 11-15 years respectively. The significant
increase in deep tillage soil porosity substantially increases soil respiration, resulting in up to 13.8 times higher
$CO_2$ emissions through increased oxidation and carbon breakdown (Reicosky et al., 1999). The lower porosities
in zero and shallow tilled soils reduces space for gas exchange, reducing soil respiration and supporting carbon
sequestration, thus increasing recalcitrant levels of carbon in soil. Mangalassery et al., (2014) found similar
porosity results using X-ray CT methods to measure the effect of tillage method on greenhouse gas emissions,
finding significantly higher porosity in tilled soil (13.6%) compared to zero tilled soil (9.6%) in the top 0-100 mm
layer. However, in deeper soil horizons, no difference could be found between tillage system. The findings in this
experiment agree with that study, showing both tillage methods did not differ significantly in the 100-200 mm
layer with lower soil porosities recorded.







4.1.3 *Penetrometer responses to tillage and traffic*

Penetrometer resistance (PR) is a useful parameter for evaluation of soil physical resistance to root growth (Otto
*et al.*, 2011). In general, trafficking had a considerable influence on soil PR in this study as depicted in fig. 8. The
greatest contrast in soil penetration resistance was between trafficked and un-trafficked soil with zero tillage
showing the highest resistance under conventional tyre pressure. Recent studies have shown that roots can exploit
pores and bypass layers of strong soil (Atkinson et al., 2020). Axial pressure from repeated trafficking in ZTCP
resulted in the highest PR values. However, root depth was less affected in contrast to STCP and DTCP. This
might explain why roots could exploit existing pore networks in undisturbed soils compared to tillage treatments.
In the middle layer examined, shallow till conventional pressure treatments suffered from a tillage pan effect
shown in Fig. 7. In fact, all trafficked zero and shallow tillage systems resulted in PR values beyond 2,000 kPa, a
threshold level which several studies show there is a reduction in root growth (da Silva, Kay and Perfect, 1994;
Lapen et al., 2004; Tormena, da Silva and Libardi, 1999). A compact zone at shallow depths is detrimental to
plant growth and crop yield in rainfed temperate climates when short term droughts occur (Campbell, Reicosky
and Doty, 1974).



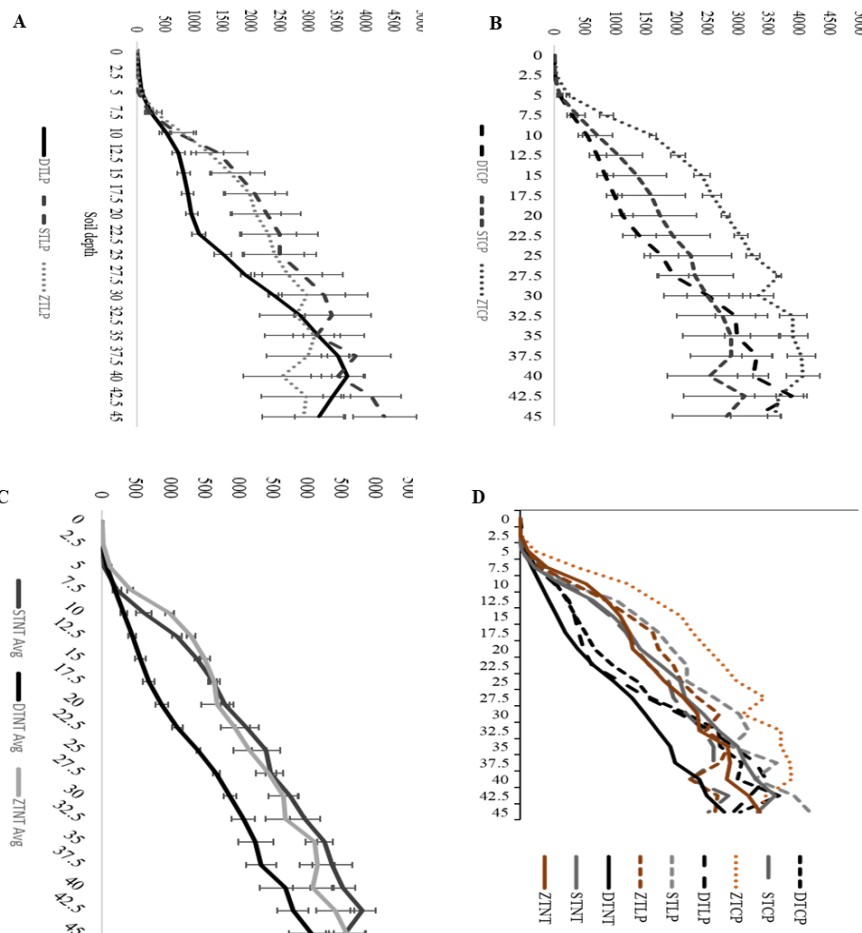

**Figure 8.** Penetration resistance (kPa) for tillage and traffic treatments at soil depths of 0 - 450 mm. X axis depicts soil depth. Y axis depicts Soil penetration resistance (kPa). Treatments represented by initials (Tillage: D = Deep, S = Shallow, Z = Zero), (Traffic: NT = No traffic, LP = Low pressure tyre, CP = Conventional pressure tyre). **A** low tyre pressure, **B** conventional tyre pressure, **C** no traffic and **D** traffic x tillage treatments combined.

4.2 *Root system architecture responses to tillage and traffic*

The 'hidden half' (i.e. roots) of plants are difficult to interpret in field studies (Lynch and Brown, 2001). A large root system is characterized by large biomass, root length and root length density (Ehdaie et al., 2010; Hamblin and Tennant, 1987). Root biomass was an important indicator of root size, showing treatment effect at anthesis compared to the tillering stage. In general, root biomass had a positive relationship with grain yield. Zero tillage treatments both untrafficked and trafficked at low pressure had greater root biomass over all shallow tillage



542 treatments and deep till trafficked at conventional pressure. Although deep tillage treatments with no traffic had

543 the highest root biomass by GS61, it did not achieve the highest yield. No significant difference in root biomass

544 was found between tillage treatments in untrafficked samples, confirming that roots are more sensitive to

545 trafficking than tillage method. The compaction effects of trafficking on soil structure exacerbated the impact on

546 rooting in general. Typically, studies report shallower rooting, increases in root diameter and decreased axial and

547 lateral rooting (Grzesiak et al., 2014). Due to the high moisture deficits depicted in (Fig S1) experienced during

548 April and May 2019, it is likely that the deeper vertical rooting in zero tillage treatments retained more moisture

549 at depth compared to other establishment methods.

550

551  Traffic significantly affected root volume, root surface area, root length and RLD in shallow tillage

552 treatments and zero tilled treatments trafficked at conventional pressure. RLD is an important parameter for

553 characterizing root growth (Doussan et al., 2006) and has been used in previous studies as a key root parameter

554 for modelling water uptake (Tinker and Nye, 2000; Javaux et al., 2013). Munos-Romero et al., (2010) and

555 Chakraborty et al., (2008) results indicate that RLD is a positive predictor of crop yield. Although RLD had a

556 positive correlation with crop yield in this study, root depth (using X-ray) displayed a much stronger relationship

557 with crop yield (fig. S3). When comparing the highest root biomass (under deep tillage with no traffic) and bulk

558 density results in the 100-200 mm layer, we found a reduction in root biomass when trafficked under conventional

559 pressure by 28% in deep tillage under conventional pressure (BD = 1.66 g cm$^{-3}$), 37% in shallow till conventional

560 pressure (1.437 g cm$^{-3}$) and 39% in zero tillage conventional pressure (1.583 g cm$^{-3}$) treatments. Colombi and

561 Walter, (2017) observed decreased shoot dry weights in pot studies by 19 and 82% under moderate (1.45 g cm$^{-3}$)

562 and high (1.6 g cm$^{-3}$) soil strength conditions. In the same study root dry weight was also reduced by 36 and 87%

563 under the same soil strength conditions. Shallow tillage had the lowest root biomass in both trafficked and

564 untrafficked treatments. Shallow tillage treatments suffered from visible horizontal fissures or "tillage pan" in Fig

565 10, causing significantly reduced rooting compared to deep tillage treatments. Moreover, a combination of <10%

566 porosity and PR reaching >2,000 kPa in the 100-200 mm layer, it is likely that roots may also have suffered from

567 anaerobic conditions due to poor infiltration rates through the tillage pan during heavy rainfall events. Conversely,

568 root impedance may have occurred during drought periods through May and June (Batey, 2009). Alameda, Anten

569 and Villar, (2012) proposed that axial growth suffers more than radial root growth. These effects of increased PR

570 and soil bulk density were observed underin the current study. However, the increase in root diameter reported by

571 several authors was not detected here (Chen et al., 2014; Lipiec et al., 2012; Tracy et al., 2012; Alameda, Anten

572 and Villar, 2012).

573

574 4.3 *2D & 3D imaging for studying root-soil relationships*

575 Due to the complexity of measuring root systems, two methods were conducted to provide comprehensive

576 analysis. Important topology (root networks) and geometrical (physical positions) characteristics of wheat rooting

577 using X-ray CT were found in this study. A strong significant relationship between RLD (WinRHIZO™) and root

578 depth (X-ray CT) was found (fig. S2) validating the suitability of image analysis methods in field studies. Further,

579 root depth showed the strongest correlation with crop yield compared to root biomass and RLD (fig. S3).



Moreover, the large environmental variance (low r number) in root relationships may have been caused by spatial
effects reported in previous studies ( Guo et al., 2020; Zhou et al., 2021). Compared to traditional 2D
WinRHIZO™ analyses, the significant difference found with *in-situ* root depth between treatments using X-ray
CT was not detected by destructive WinRHIZO™ analysis (i.e., it involves the washing of soil from root material,
thus losing important architectural data). Destructive root analysis showed evidence of superior rooting properties
under deep tillage treatments (e.g., root length density and root volume). Visualizing important behaviors of wheat
rooting in field scale trials, highlights the importance of root depth to sustain high yields in drought conditions.
Figure 9 depicts significantly longer root length in zero tillage treatments compared to trafficked deep and shallow
tillage, with trafficked treatments roots were generally confined to the top 0-50 mm of soil. In general, root length
rarely surpassed 100 mm in depth. This was partly due to insufficient resolution available with the X-ray CT
scanner to capture finer root materials (Pfeifer et al., 2015).
In general, both root analysis methods showed agreement in the results. Zero tillage treatments had
significantly deeper rooting over shallow tillage and deep tillage trafficked treatments. Using the WinRhizo™
method, untrafficked deep tillage treatments showed superior root length. Similar disagreements in findings
between methods could be explained by the difference in methodology between the two imaging approaches as
X-ray CT is 3D and scans roots in soil whilst, WinRhizo™ is 2D and scans washed roots  (Tracy et al., 2012).
Root volume and surface area were also examined using X-ray CT. In contrast to the WinRhizo™ analysis, no
significant differences could be detected between treatments. The root volumes obtained by the WinRhizo™ were
much greater than the volumes attained from the X-ray CT scan. The difference can be attributed by much clearer
contrasts between air and root material with the destructive method compared to limitations with resolution and
density differences between soil, root and organic materials (Mooney et al., 2012) in the X-ray CT scan images.

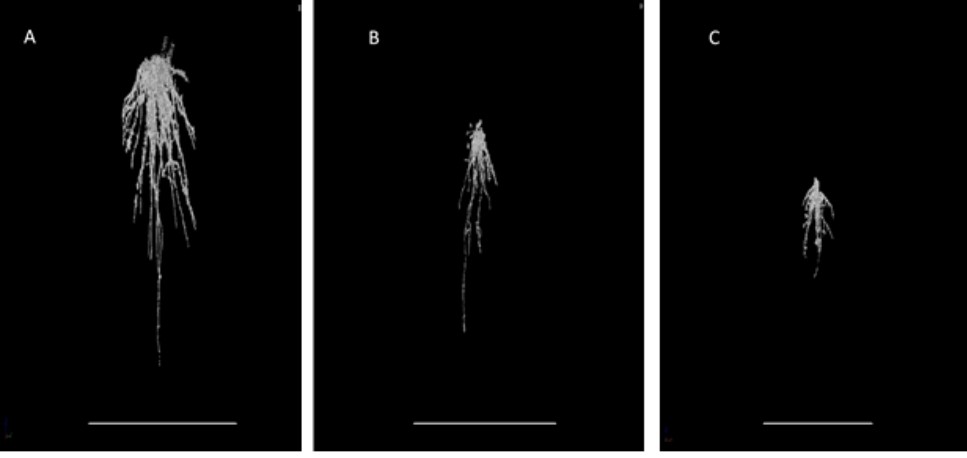


**Figure 9.** Root system architecture of winter wheat during anthesis for (**a**) Deep tillage no traffic, (**b**) Zero tillage
low tyre pressure and (**c**) deep tillage conventional tyre pressure. (a) and (b) showed significantly longer root
length on the primary axis compared to (c) deep tillage trafficked treatments. Scale bar = 70 mm.









4.4 *Traffic and tillage effects on rooting and crop yield*
In the present study, it was found that long term zero tillage plots under low tyre pressure increased yield by up
to 0.772 Mt ha⁻¹ compared to the deep tillage conventional tyre pressure treatments. All zero tillage treatments
yielded over 11 Mt ha⁻¹ compared to deep and shallow tillage treatments (10.71 Mt ha⁻¹mean). Evidence using
data collected from the X-ray CT scans showed deeper vertical rooting in zero tillage plots compared to shallow
and deep tillage treatments (Fig. 9). Coupled with deeper rooting, zero tillage no traffic treatments had
significantly lower bulk density than deep tillage conventional pressure plots. Munoz-Romero et al., (2010)
reported a yield increase of 0.5 Mt ha⁻¹ in zero tillage compared to conventional tillage which was associated with
greater water use and increased water use efficiency, similar to (Chakraborty et al., 2008). Improvements in
moisture retention, soil pore structures and reduced soil compaction under zero-tillage, may also have contributed
to a yield increase over conventionally tilled treatments.
It is possible that the lower levels of porosity found in zero tillage aided with water retention during drought
periods on the highly sandy soil in this trial. Coupled with the development of vertically oriented soil structural
characteristics attributed to earthworm activity and old root channels (Fig 7), the zero tillage treatments may also
have had increased access to water by roots at lower soil horizons. Indeed, biopores benefit root growth by altering
the surrounding chemical, physical and biological properties of soil ( Stroud et al., 2017; Banfield et al., 2017).
Thus providing macropore pathways with lower mechanical resistance in which deeper rooting preferentially
grow towards (Zhou et al., 2021).   In contrast, deep cultivation created a porous structure which has shown to
increase  respiration of aerobic microorganisms, improving the flow of air and water thus increasing $CO_2$
emissions ( Mangalassery et al., 2014). Crop yield was influenced less in zero tillage treatments by trafficking
than the other tillage treatments. The lower sensitivity to compaction in zero tillage is attributed to an elastic
behavior or increase in bearing capacity, with soil acquiring similar structural properties to grassland soil (Ehlers
and Claupein, 1994).







## 5 Conclusion

The results from this research highlight the importance of traffic management for improving crop productivity. Physical and visual implications of soil compaction on the soil profile were demonstrated in this study, signifying the implications of tyre pressure on root growth. High tyre pressure significantly reduced root development in all tillage treatments. However, deep, and shallow tillage systems were more influenced by compaction with roots confined to the top 0-60 mm thus, reducing primary vertical rooting and inhibiting roots access to deeper soil moisture reserves. The highly significant impact on crop yield was highlighted by the strong relationship between root depth and crop yield. The visible effects of trafficking on the soil profile depicted through X-ray CT, provides evidence of the damage modern farm machinery can cause for root resource capture, leading to potential increased drought stress and yield loss in crop production. This long-term trial site has shown that zero tillage does not affect root growth, in fact, reduced bulk density, improved grain yield and rooting depth significantly through deeply connected vertical soil pore fissures created by earthworms and old root channels. These findings suggest that scientists and farmers should focus on designing improved zero tillage cropping systems, managing field trafficking protocols. Furthermore, this research shows that the combination of X-ray CT scanning along with traditional destructive methods provide a robust method for assessing in field rooting for future crop breeding initiatives and soil management practice. This research concludes that little differences were found between deep tillage and zero tillage methods in the absence of traffic in terms of overall physical root growth. However, in abundance of biopores and increased soil bearing capacity to withstand machinery traffic in in zero tillage systems increased rooting depth and moisture retention during the growing season.

**Supplement.** The supplement related to this article is available in a separate word file as per submission.

**Author contributions.** KMc and ST conceived the experiment. DH & MH carried out sampling and soil analysis. DH processed and analysed all samples. DH analysed and interpreted the data and wrote the manuscript. All authors contributed to the data, providing interpretation and comments to the manuscript.

**Competing interests.** The authors declare that they have no conflict of interest

**Acknowledgements.** This research took samples from a long term tillage and traffic experiment site in Harper Adams University. Long term crop rotation treatments at the Large Marsh site on the University grounds are managed and maintained by the agricultural staff at the university.



674

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
