# Peer review of "The effect of tillage depth and traffic management on soil properties and root development during two growth stages of winter wheat (*Triticum aestivum L.*)"

_SOIL, 2021_

## Author Response (AR1)

**Point-by-point reply to reviewers comments and changes made**

**Reviewer 1**

Dear Editor and authors.

I found the study quit interesting and well-worth to be publish. It has been used a proper and very interesting methodology for the roots system analysis.

*Authors:* **On behalf of the authors, we our very thankful for the reviewers positive feedback and interest in the study.**

I had some problems to understand several results in the figures. Mainly.. In the X-axis it is difficult to see well the combination of the treatments because is like a continuous line.

Secondly and due to the 9 combinations would be more appropriate to use the same secuency of those combination in all the figures.

I suggest a better way to present those figures.

Also it is important was indicate the meaning of the bars in the legend of the figure.

It is difficult to see in the figures the effect of the tillage and the effect of the trafficking individually.

*Authors:* **Point accepted. We agree that the figures require further attention to detail and will correct same. We will ensure that the next draft is consistent with treatments in the figures.**

**Changes made**: Some figures were removed and variance of analysis tables created to improve the visibility of the treatment effects. A table was added for the bulk density, penetrometer resistance and WinRHIZO root results.

Regarding the abstract I would suggest give some numbers mora than qualify sentences.

*Authors:* **Understood. We will add the statistical significance of the results into the abstract.**

**Changes made:** P values were added to the abstract in the latest marked up version.

The introduction and the methodology are quite OK.

THe resuts need better way to understand figures and the effect of the factors.

*Authors:* **We have prepared the results in a better format for the next draft which will hopefully improve clarity of the results.**

**Changes made: Tables and figures have been improved with appropriate legends and consistency.**

The discussion is quite long and also includes some data that could go to results.

Finally the discussion should give a clear picture of the two factors. Tillage and traffiking. Two sections.

**Changes made:** The discussion was split between tillage and traffic effects. Excessive detail was removed from the discussion section and shortened.

*Authors:* Thank you for your constructive feedback. We will amend for the next draft.

IN the way is written it looks again a show of results with some discussion. NO clreaer picture of the Tillage and trafkiking.

Best regards.

**Reviewer 2**

The manuscript addresses the influence of tillage and traffic management on soil properties and root development. It has a good level of detail and it fits nicely with the topic of the journal. The overall structure of the paper is good, however, there are some concepts that need to be more clearly defined.

*Authors*: On behalf of authors, I would like to thank Referee 2 for his/her time, helpful and constructive comments on our work.

Abstract: I think it should be better stated which factor is more important management wise and its consequences.

*Authors:* Suggestion accepted. The managment systems between traffic and tillage are highly inter-related We will improve the message in the next draft to clearly represent managment approaches and consequences.

**Changes made:** I have made it clearer that trafficking had more of an influence on crop yield and root growth. P values were added into the abstract to highlight the main findings.

Introduction: The purpose of the paper needs to be rephrased as it's not clear they are assessing the influence of trafficking. The cultivation and traffic management description should be moved to methodology.

*Authors:* Thank you for your feedback. A full review of the introduction will be conducted to sharpen the focus on both traffic managment strategies.

**Changes made:** The purpose of the paper has been rephrased in the marked up version.

M&M: In my opinion, it would be important to add a description of the tillage history of the site. The traffic regimes need to be better defined specifying or justifying how the combination of deep tillage without trafficking is possible. It appears the concept of controlled traffic farming (CTF) is not mentioned again throughout the paper. Regarding the Soil Moisture Deficit Model, what are the inputs of the model? only weather parameters?. Why did the authors choose to transform all the data to be normally distributed instead of using a non-parametric test?

*Authors:* Agreed, the history of the site is an important component of tillage and trafficking trial work. We will add more details in M&M to improve the visibility of the treatments conducted. The inputs of the model include max and min temperatures, rainfall, windspeed (M/s) and sunshine hours per day. We chose to transform data that did not show normality. Each dataset was tested for normality before conducting any statisitcal analysis. Parametric tests have more statisitcal power, testing the mean of each dataset.

**Changes made:** The marked up version includes the site history and more detail on the trafficking regimes and how they were conducted. The inputs of the soil moisture deficit model are described in the methods section and reference included if further details required.

Results: In general, figures need substantial revision. Legends should indicate the meaning of the treatments as in figure 8 and the same format (sequence of treatments) should be kept throughout. Also, the description of the results seems confusing at times.

*Authors:* Agreed - We will revise the figures and improve on the consistency such as formatting.

**Changes made:** I removed figures from the result section and replaced them with linear variance of analysis tables to improve the message and provide a clear picture of the treatment effects. Figure 8 was improved with same sequence.

Discussion: The discussion is too long and also it's a bit redundant as it includes some data that should go to results or be removed. I think it could be useful to split it into tillage or traffic management effects.

*Authors:* We will split the discussion section into two sections and thank you for your helpful feedback.

**Changes made:** The discussion was split between traffic and tillage effects on soil properties and roots where possible. Excessive detail was removed from the discussion as shown in the marked up version of the manuscript.

Conclusions: they should state better what the recommendations are from an agronomic point of view and also the reflection of whether it would be really possible to leave traffic-free zones and what the advantage would be instead of establishing Low-pressure tyre zones?

*Authors:* Point taken and we will amend.

**Changes made:** Recommendations and further investigations are included in the marked up version.